# Comprehensive Genomic Analysis and Expression Profiling of Diacylglycerol Kinase (*DGK*) Gene Family in Soybean (*Glycine max*) under Abiotic Stresses

**DOI:** 10.3390/ijms20061361

**Published:** 2019-03-18

**Authors:** Kue Foka Idrice Carther, Toi Ketehouli, Nan Ye, Yan-Hai Yang, Nan Wang, Yuan-Yuan Dong, Na Yao, Xiu-Ming Liu, Wei-Can Liu, Xiao-Wei Li, Fa-Wei Wang, Hai-Yan Li

**Affiliations:** College of Life Sciences, Engineering Research Center of the Chinese Ministry of Education for Bioreactor and Pharmaceutical Development, Jilin Agricultural University, Changchun 130118, China; kuefokaidricecarther@yahoo.com (K.F.I.C.); stanislasketehouli@yahoo.com (T.K.); yenan18@163.com (N.Y.); yyh199873@163.com (Y.-H.Y.); wangnanlunwen@126.com (N.W.); yydong@aliyun.com (Y.-Y.D.); yaona1103@163.com (N.Y.); xiuming1211@163.com (X.-M.L.); liuweican602@163.com (W.-C.L.); xiaoweili1206@163.com (X.-W.L.)

**Keywords:** soybean, diacylglycerol kinase, phosphatidic acid (PA), abiotic stress, gene family lipid signaling

## Abstract

Diacylglycerol kinase (DGK) is an enzyme that plays a pivotal role in abiotic and biotic stress responses in plants by transforming the diacylglycerol into phosphatidic acid. However, there is no report on the characterization of soybean *DGK* genes in spite of the availability of the soybean genome sequence. In this study, we performed genome-wide analysis and expression profiling of the *DGK* gene family in the soybean genome. We identified 12 *DGK* genes (namely *GmDGK1-12*) which all contained conserved catalytic domains with protein lengths and molecular weights ranging from 436 to 727 amino acids (aa) and 48.62 to 80.93 kDa, respectively. Phylogenetic analyses grouped *GmDGK* genes into three clusters—cluster I, cluster II, and cluster III—which had three, four, and five genes, respectively. The qRT-PCR analysis revealed significant *GmDGK* gene expression levels in both leaves and roots coping with polyethylene glycol (PEG), salt, alkali, and salt/alkali treatments. This work provides the first characterization of the *DGK* gene family in soybean and suggests their importance in soybean response to abiotic stress. These results can serve as a guide for future studies on the understanding and functional characterization of this gene family.

## 1. Introduction

Drought stress represents the most important limiting factor that affects plant growth and development, and severely reduces worldwide plant production [1,2]. Many studies were conducted on plants using biotechnology methods to improve their abilities in adaptation to water shortage stress.

Soybean (*Glycine max*) is a major oilseed crop and an important source of vegetable oil and protein for humans and animals worldwide. Soybean is one of the most important grain legumes, ranking among the top five worldwide major crops, generating 18–22% of essential sources of oil and 35–40% of protein nutrients and minerals. It is primarily grown in the tropics and semi-arid tropics (http://faostat.fao.org/). Many studies were conducted to improve the production of soybean under different environmental stress conditions [3,4,5]. In fact, soybean is very sensitive to drought stress because of its high water requirement [6]. We believe that the use of plant biotechnology is essential for the management of those genotypes which could efficiently exploit the use of available water.

In plants, the reaction processes against biotic and abiotic stress involve important signaling molecules, such as phosphatidylinositol lipids (PPIs), phosphatidic acid (PA), diacylglycerol (DAG), and some lysophospholipids, which are activated in response to different stresses including drought, saline, alkaline, and cold conditions [7,8]. Among these molecules, PA has the fastest and most transient signal production, playing an essential role in plant stress responses as a second messenger in plant signal transduction, by enrolling cytosolic proteins to their assigned membranes. PA is immediately generated from the phosphorylation of diacylglycerol (DAG), catalyzed by the diacylglycerol kinase (DGK) enzymes [6]. The *DGK* gene family was extensively investigated and characterized based on the whole-genome sequences of several plants, and the genes were grouped into three clusters (clusters I, II, and III) [9,10]. For example, in *Arabidopsis thaliana* there are seven *DGK* genes, (*AtDGK1* to *AtDGK7*), while there are eight *DGK* genes in rice (*OsDGK1* to *OsDGK8*), seven genes in maize (*ZmDGK1* to *ZmDGK7*), and eight *DGK* genes in *Malus domestica* (*MdDGK1* to *MdDGK8*). In *Arabidopsis*, *AtDGK1* was cloned and it was expressed in the roots, leaves, and shoots, but not in the siliques and flowers [11]. Later, *AtDGK2* was also cloned and its transcripts were expressed in most components of the plant, whereby expression was triggered by chilling at 4 °C, pointing out the participation of *DGK* signal transduction in plant cold stress response [12]. Afterward, knock-out mutants facilitated the analysis of *AtDGK1* and *AtDGK2*, revealing their participation in the cold-responsive transcriptome of *Arabidopsis* [13]. A recent study in apple discovered the expression of both *MdDGK4* and *MdDGK8* transcripts in response to salt and drought stress, respectively [14]. Thus, *DGKs* could play a crucial role in several plants abiotic stress responses such as cold stress, drought stress, and salt stress, due to the strategic positioning between two ultimate lipids (DAG and PA). However, few studies were conducted on *DGK* genes, despite their significant role in lipid signaling in plants. Therefore, further studies are required to elucidate their specific functions in soybean plants.

*Glycine max* is a dicot which can be consider as a plant model in genomic research. Soybean whole-genome identification [15] provided insights into new methods for soybean gene investigations, thus facilitating the discovery of new abiotic-related genes, which might lead to the elucidation of a gene evolution model and positioning of their functions in plant. However, the knowledge of the molecular functions of *DGK* genes remains primitive in soybean, and this work is the first study on the *DGK* gene family in soybean plants.

In this study, we inventoried 12 *DGK* genes for the DGK family in the soybean genome, and we analyzed the gene structure, the subcellular localization, the evolution, the protein conserved domains (the cis-motif promoters and others to postulate information on the evolutionary relationships), and the molecular features of *Glycine max DGK* genes through bioinformatics analysis. Then, we validated the bioinformatics results by performing real-time PCR in two major plant tissues (leaves and roots) under PEG, salt, alkali, and salt/alkali stress to determine the contribution of the soybean *DGK* genes in such circumstances through the analysis of their expression patterns in both tissues. Thus, we performed a genome-wide identification of *DGK* gene family members in *Glycine max* and their transcript expressions under drought stress. The purpose of this study was to improve the understanding of *DGK* gene involvement in soybean response mechanisms to abiotic stress. This work may provide basic knowledge for future studies to determine the functions of *DGK* genes in soybean.

## 2. Results

### 2.1. Genome-Wide Identification and Chromosomal Distribution of DGK Gene Family Members in Soybean

We identified a total of 12 *DGK* genes that we designated as *GmDGK1–12*, and we grouped them in a table with their different identifiers and their open reading frames (ORFs) lengths, ranging from 1185 bp (GmDGK1) to 2184 bp (*GmDGK12*).

The characterization of the soybean *DGK* gene polypeptides showed long sequences with 436–727 aa, molecular weights (MWs) of 48.6–80.93 kDa, and protein isoelectric points of 5.29–7.61 (Table 1). All 12 genes were located on six of the 20 soybean chromosomes (Chr); *GmDGK1* and *GmDGK7* were found on Chr 13, *GmDGK2* and *GmDGK12* were found on Chr 5, *GmDGK3* and *GmDGK8* were found on Chr 12, *GmDGK4*, *GmDGK5*, and *GmDGK7* were found on Chr 6, and *GmDGK10* and *GmDGK7* were found on Chr 17 (Figure 1).

### 2.2. Multiple Sequence Alignments and Sequence Characterization of the Soybean DGK Genes

The multiple cluster alignment of the soybean diacylglycerol kinase catalytic domain (DGKc) was represented in three phases. Figure 2A shows DAG/phorbol ester (PE)-binding domain 2, Figure 2B shows DAG/PE-binding domain 1, and Figure 2C shows the diacylglycerol kinase accessory (DGKa) domain. The alignment revealed that, except for *GmDGK2*, *GmDGK8*, and *GmDGK9*, almost all *GmDGKs* conserved the ATP-binding motif, represented by a sequence of GXGXXG in their catalytic domain (DGKc), as shown in the marked area of Figure 2A. As in other studied plants, cluster I also featured the DAG/PE-binding domain (C1 domain, PF00130), which harbors the sequence HX14CX2CX16–22 CX2CX4HX2CX7C and the DAG/PE-binding domain HX18CX2CX16CX2CX4HX2CX11C [16]. This DAG/PE-binding domain holds a preserved 15 amino-acid (aa) extension, and this domain was, therefore, named the “extended cysteine-rich domain” (extCRD) [17].

However, all plant DGKs have a universal framework in cluster I: (YT-upstream basic region-VP)–(3 aa)–(DAG/PE-BD-1)–(12 aa)–(DAG/PE-BD-2/extCRD-like)–(~130 aa)–(DGKc/DGKa domain). Hence, it is probable that the extCRD, in combination with the preserved catalytic domain, is indispensable for the sole operative enzyme of DGKs [18]. Cluster I alignment of the *DGK* genes of soybean remained almost the same as in *Arabidopsis*, rice, and apple with some slight modifications (Figure 3). For example, even if the conservation of the KA, AR, and KK residues was removed from the basic region, YT and VP residues remained as boundaries which were conserved in *Arabidopsis*, rice, apple, and soybean, except that in *MdDGK1*, in which Y and T were substituted by F and A respectively. While in *GmDGK2*, Y was replaced by A; and finally, V was replaced by I in *GmDGK11* and *GmDGK12* (Figure 3).

### 2.3. Phylogenetic Analysis of Soybean DGK Gene Family

A phylogenetic analysis was established based on apple, *Arabidopsis*, rice, and soybean using protein successions [19,20]. The obtained tree showed three different clusters, distributed as follows: three genes, *GmDGK2*, *GmDGK11*, and *GmDGK12*, were found in cluster I; four genes, *GmDGK5*, *GmDGK6*, *GmDGK7*, and *GmDGK10*, were found in cluster II; and five genes, *GmDGK1*, *GmDGK3*, *GmDGK4*, *GmDGK8*, and *GmDGK9* were found in cluster III. The phylogenetic tree indicated that the clustering of *GmDGK*s was narrow compared to the organization of the *DGK* genes from the dicot *Arabidopsis* and the monocot rice. Three paralogous gene pairs were also identified in the soybean genome: *GmDGK1/GmDGK3*, *GmDGK7/GmDGK10*, and *GmDGK11/GmDGK12* (Figure 4).

### 2.4. Structures and Protein Motifs of Soybean DGK Genes

The exon–intron structure of each gene of interest in the soybean *DGK* genes in comparison to *Arabidopsis DGK* genes was determined by the matching of their genome sequences, in an effort to have greater visibility of their evolution (Figure 5). Furthermore, the organization of the coding sequences showed that all the soybean *DGK* genes contained seven exons in cluster I and 12 exons in other clusters (II and III), except for *GmDGK9* in cluster III, which had 13 exons (Appendix A). This proximity between the exon–intron associations suggested that gene duplication occurred throughout the soybean *DGK* gene family over evolutionary time [21].

### 2.5. Protein Domains in Soybean DGKs

The mapping of the entire GmDGK protein domains was executed with the DOG 2.0 software (Figure 6). The diagram indicates different distributions of the functional domains of GmDGK proteins, based on the conservation of the macro protein domain throughout the evolution of all three clusters. Despite all 12 GmDGK proteins containing one catalytic domain (DGKc) and one accessory domain (DGKa), only cluster I (*GmDGK2*, *GmDGK11*, and *GmDGK12*) had the DAG/PE-binding motif and transmembrane regions (C1). However, the localization of the functional domains belonging to the same clusters was almost identical. For example, in cluster II, the DGKc domains of *GmDGK4* and *GmDGK9* began at the 36th aa, while the DGKc domains of *GmDGK7* and *GmDGK10* in cluster III started from the 83rd aa (Figure 6). Also, the position of the DGKa domain was between the 176th and the 406th aa in cluster I, while it existed between the 275th and 457th aa in cluster III.

### 2.6. Expression Analysis of GmDGKs under Abiotic Stress

Quantitative RT-PCR analysis was performed to evaluate *GmDGK* transcript expression levels in leaf and root tissues under PEG, saline, alkaline, and saline/alkaline treatments. The qRT-PCR results revealed that the expressions of *GmDGK1*, *GmDGK5*, *GmDGK6*, *GmDGK8*, and *GmDGK9* were lower in both root and leaf tissues, especially for *GmDGK5* and *GmDGK6*, which were undetectable in both tissues.

During the PEG treatment, the *GmDGK10* transcripts recorded a dramatic 22.5-fold upregulation at 9 h in the leaf tissues, before decreasing to the lowest amount at 24 h, whereas a 2.5-fold change was the average of other *GmDGK*s (Figure 7). In roots, both *GmDGK7* and *GmDGK10* in cluster II were approximately 2.8-fold upregulated at 3 h under PEG application, while the highest upregulation (4.5-fold) was shown by *GmDGK2* in cluster I at 12 h (Figure 8).

The salt treatment results revealed that, except for *GmDGK2*, all *GmDGK*s were significantly upregulated at the end of the treatment (9 h and 12 h) in roots, and only *GmDGK10* showed a record 5.3-fold increase in expression at 6 h. On the other hand, in leaf tissues, *GmDGK8* and *GmDGK9* in cluster III had higher expressions 6.6- and 9.8-fold, respectively) after 1 h under salt treatment (Figure 7).

The trend of results during alkaline stress was almost similar for all *DGKs* in root tissues, with a punctual upregulation for 1 h and a gradual decrease throughout the duration of the treatment, especially for genes in clusters I and II. However, in leaf tissues, *GmDGK2* and *GmDGK10* showed the highest upregulation at 9 h and 12 h, with 6.2- and 8.4-fold increases, respectively.

The combined salt/alkaline stresses induced upregulations in all *DGK* gene expressions for both leaves and roots at different time points. Whereas *GmDGK*s in cluster III were slightly regulated, we noted that the *GmDGK*s in clusters I and II were strongly upregulated in both tissues with a maximum fold change of 4.3 for *GmDGK2* in leaf tissues at 9 h of stress application (Figure 7). A similar study was conducted on maize (*Zea mays*), and it was reported that the relative expression levels of *ZmDGK7* were prominent in many tissues under salt stress treatments [16]. These results indicated that, in *Glycine max*, various isoforms of *GmDGK*s could react differentially to several abiotic stresses, and that the upregulation of all *GmDGK*s under PEG, saline, and alkali treatments suggested a specific role of *GmDGK*s in abiotic stress responses.

### 2.7. Expression Profiles of Soybean DGK Genes in Different Tissues

The tissue-specific expression of all *DGK* genes in soybean was determined to compare their expression levels in both root and leaf tissues; thus, qRT-PCR was monitored for all 12 genes on untreated plants (*T* = 0 h). The results showed that, except for *GmDGK5* and *GmDGK6*, which were excluded from our examination due to their non-expression in leaf and root tissues (with a very low expression of *GmDGK6* in roots) for several assays, the expression levels of every other genes were ubiquitous in the leaves and roots; generally, all genes showed noteworthy expressions in both tissues. However, there were no significant contrasts among *DGK* gene expression levels in both leaf and root tissues (Figure 9); nonetheless, *GmDGK2*, *GmDGK4*, *GmDGK7*, and *GmDGK11* transcript levels were higher in both tissues. These important indices with regard to similarities and differences among the two tissue expressions suggested that *GmDGK*s are involved in different steps during root and leaf development [22].

### 2.8. Prediction of Subcellular Locations for DGK Proteins in Soybean

From 1989 to 1992, studies were conducted on the activity of *DGK* genes and their subcellular localization in plants. *DGK* was previously found in the nucleus [23], the plasma membrane, and the cytoskeleton [24]; however, DGK activity was also reported in the chloroplast [25]. The predictions of the subcellular localization of soybean DGK proteins, using results from the DeepLoc1.0 online tool sets for eukaryote cell protein localization, suggested that all cluster I GmDGKs (GmDGK2, GmDGK11, and GmDGK12) are mostly located in the plasma membrane and endoplasmic reticulum, while cluster II GmDGKs (GmDGK1, GmDGK3, GmDGK4, GmDGK8, and GmDGK9) and cluster III GmDGKs (GmDGK5, GmDGK6, GmDGK7, and GmDGK10) were located in the cytoplasm and nucleus. GmDGK10 was predominantly detected in the nucleus (Appendix A).

### 2.9. Cis-Acting Elements in the Promoter of the Soybean DGK Genes

The 1.5-kb cabled upstream site of all *GmDGK* promoters from the starting codon was drawn from the soybean genome, and we identified all cis-acting regulatory components for each single promoter sequence. From these observations and analyses, we selected some indicative components for expression investigation. Thus, the hosting of cis-regulatory units in all promoters of the soybean *DGK* gene family, such as *ABRE* (drought stress responsive), *GT1*, (light-regulated gene), *RHe* (hairy specific element), *MYB2* (water stress-responsive), and *DRE* (cold stress-responsive), proves that the *DGK* genes could play a primordial role in soybean response under abiotic stresses (Figure 10).

## 3. Discussion

The present work involved a complete overview of the *DGK* gene family in soybean. We performed in silico identification and elucidation of the potential role of the diacylglycerol kinase genes in *Glycine max* using bioinformatics resources. Thus, we carried out bioinformatics analysis of the *DGK* gene family members, including soybean chromosome gene structure, gene sequence characterization, phylogenetic relationships, conserved motifs, evolutionary patterns, cis-regulatory elements, and protein subcellular localization, with the expression analysis of the *DGK* gene family under abiotic stress.

Diacylglycerol kinases (DGKs or DAGKs) are kinases that can use ATP as an energy generator to catalyze the phosphorylation of diacylglycerol (DAG) to phosphatidic acid (PA) [26]. There are two categories of DGKs in eukaryote cells. On the one hand, inactive DGKs with a lower cellular activity are stimulated by the metabolic signaling of inositol phosphate (IP_3_), which produces phosphoglycerols from DAGs. On the other hand, active DGKs, which have greater cellular activity and catalyze the production of PA, play a key role in PA signaling [27]. In silico identification was used for the functional predictions of the *DGK* gene family in *Arabidopsis*, apple, rice, and maize, and the primer background was used for further studies on the *DGK* gene functional analysis in these plants under drought stress. The upregulation of *DGK* genes in maize under drought, salt, and cold stresses and their obvious downregulation during salt stress treatment [16] first revealed DGK activity under drought stress. Later, researchers discovered the modulating role of *DGK* genes in root development under salt stress in *Arabidopsis* [13], the higher response of the *DGK* genes under drought stress treatment in apple [14], the expression of both *AtDGK1* and *AtDGK2* in the *Arabidopsis* cold-responsive transcriptome [28], and several repressions observed on *DGK* genes under various stresses in other plants (barley and wheat). These findings suggest *DGK* genes as pivotal in plant stress responses, and they promote further investigation on the relationship between abiotic stresses and the paired phospholipase C (PLC)/DGK pathway in plant response [29].

This curiosity prompted our study concerning the roles of *DGK* genes in soybean, and, for this study, we chose soybean William 82 as a model because of three factors. Its whole genome was sequenced in previous study [15], it was identified as having an efficient drought-response tolerance [30], and its tolerance could be improved [31].

At first, we identified 12 *DGK* genes located on six soybean chromosomes (Chr 4, Chr 5, Chr 6, Chr 12, Chr 13, and Chr 17) (Figure 1). We also identified seven *Arabidopsis DGK* genes [32], eight in rice [33], and eight in apple [14] to compare the evolutionary patterns of the *DGK* gene family among monocots and dicots. We found that the clustering of *GmDGKs* was narrow compared to the organization of the *DGK* genes from the dicot *Arabidopsis*, and from the monocot rice. The wide diffusion of *DGK* genes in soybean compared with *Arabidopsis* point out its positive selection and dynamic expansion in soybean. The phylogenetic analysis gave rise to three distinct clusters (cluster I, II, and III), whereby the three *GmDGK*s in cluster I are the most complex *GmDGK*s in soybean, and they are very similar to other plant genes in cluster I such as rice, apple, maize, and *Arabidopsis*. Also, by comparing the *DGK* gene structures between soybean and *Arabidopsis* regarding conserved domains and their exon/intron organization, we found that all *DGK* genes in cluster I had the same exon number (seven), and, in clusters II and III, except for *GmDGK9* and *AtDGK4* (10 exons) and *AtDGK7* (nine exons), all other genes had 12 exons. This similarity between genes of different species suggests that they come from the same ancestor and that the *DGK* genes were strongly affected by the repetitive phenomenon of DNA duplication during the evolution process over time [34].

Drought stress is one of the most limiting factors that inhibit plant growth and development, and reduce plant production [1,2]. Many studies were conducted on plants using biotechnological methods to improve their ability in adaptation to water scarcity all over the world, which is the most limiting factor in agricultural production. Several studies on plant *DGK* genes, such as in *Arabidopsis*, apple, and rice, were conducted under abiotic stresses, since GenBank numbers were first described for *DGK* gene homologs ranging from tomato (AW035995) to maize (AY106320), wheat (BT009326), grape (CB981130), and apricot (*Prunus armeniaca*; CB821694) to mention a few. This investigation established that *DGK* genes are widely distributed in plants and play a positive role in plant abiotic stress responses. A better understanding of their functions could be a valuable asset for a better understanding of PA pathway regulation.

Expression patterns of the 12 *GmDGKs* were determined using qRT-PCR to identify their current transcript levels in two important tissues (leaves and roots) which host the majority of processes of plant defense against environmental stress [35], with untreated plants as the control samples (0 h). We found that, except for *GmDGK5* and *GmDGK6*, which were not expressed, the other *GmDGK*s were notably expressed in leaf and root tissues, thus marking their transient activities in plant cells, with overall expression higher in leaf tissues. This suggests the regulatory role of DGK enzymes in the PA pathway during root and leaf development, explaining their higher activities in the plasma membrane [22]. Therefore, according to our results with these four stress treatments, we can provide a basis for further investigations into *GmDGK* functions according to cluster members [32]. The second qRT-PCR was employed on the treated plants, and we determined all *GmDGK* transcript levels in leaf and root tissues after PEG, saline, alkaline. and saline/alkaline treatments. Here, our results revealed that the *GmDGKs* (*GMDGK2*, *GmDGK11*, and *GmDGK12*) in cluster I were over-expressed compared to other *GmDGKs*, whereas *GmDGK4*, *GmDGK7*, and *GmDGK10* were also slightly upregulated, and *GmDGKs* in cluster III were less expressed in reaction to abiotic stresses, while *GmDGK8* and *GmDGK9* were over-expressed in root tissues during the alkaline and salt/alkaline treatments. These variations in *DGK* gene expression patterns in response to drought stresses reflected their metabolic activities in both leaves and roots [36]. The up- and downregulations of *DGK* genes throughout the PEG application revealed the potential role of *GmDGK*s in the soybean water-deficit response. Similar trends were found for several metabolites in various plants such as tomato [37], pea [38], and barley [39]. The upregulation of all *GmDGK*s in root tissues in response to salt treatment suggests their promoting roles in lateral root development under salt stress in soybean [28]. Finally, the alkaline and salt/alkaline treatments revealed significant upregulations in the expression of all *GmDGK* transcripts, which is proof of their influence on the NaCl and NaCO_3_ content of metabolites and metal elements in plants, and points to the potential role of DGK enzymes in the regulation of plant nutrient status [40].

However, the wide localization of cluster I *DGK*s in the plasma membrane suggests that cluster I *GmDGK*s might play a transitioning role between the extracellular medium and the intracellular medium as transmembrane proteins [41], since in mammal cells, *DGKθ* and *DGKα* were also located in the plasma membrane, and could move from the cytosol to the plasma membrane [42], and from the cytoplasm to the plasma membrane [43]. The prediction of *DGK* proteins in several key organelles of the plant cell indicates that *DGK* enzymes could actively take part in cellular metabolism during plant development and abiotic stress response [10].

Finally, the identification of the cis-elements in the promoter regions of soybean *DGKs* revealed various cis-motif regulator factors similar to *MBS* (MYB-binding site), *ABRE* (ABA-response factors) [44], *LTR* (low-temperature-responsive) factors [44,45], and *HSE* (heat shock elements) [46], suggesting that *GmDGK*s could cope with several abiotic stress responses in soybean plants.

## 4. Materials and Methods

### 4.1. Localization and Identification of the Soybean DGK Genes on Chromosomes

The census of the *DGK* gene was done with an online search. We used the full name of “diacylglycerol kinase” to find soybean *DGK* sequences on SOYBASE (https://soybase.org). Then, we BLASTed all soybean *DGK* sequences on the database (http://www.phytozome.org/) and we used gene identifiers (IDs) found on Phytozomev9.1 as the keywords in the browser tool on the NCBI website (http://www.ncbi.nlm.nih.gov/Structure/cdd/wrpsb.cgi) to confirm their positions on the chromosomes. Furthermore, the *DGK* gene sequences of *Arabidopsis thaliana* were collected on the online *Arabidopsis* platform Resource (The Arabidopsis Information Resource) TAIR 10.0, while rice *DGK* sequences were found on the Rice Genome Annotation Project (RGAP), and finally, we recovered the locational information of every single putative gene from Computational Science Web Assets (http://genomics.research.iasma.it/gb2/gbrowse/apple/). Then, we used Microsoft PowerPoint 2013 to sketch all *DGK* genes onto their respective chromosomes.

### 4.2. Multiple Sequence Alignments and Sequence Characterization of the Soybean DGK Genes

The soybean *DGK* gene characterization was implemented by analyzing their structures and constructing their phylogenetic tree. The multiple sequence alignments of DGK protein sequences were established with the DNAMAN (https://www.lynnon.com/pc/alignm.html) program, using the default parameter mode. The evolutionary relationship among plant *DGK* genes was found by collecting the protein sequences of four plants (*Arabidopsis*, rice, soybean, and apple) from the NCBI protein database. Then, we used the integrated MUSCLE alignment program in MEGA6.0 software to align all protein sequences, using the neighbor-joining (NJ) method and 1000 replications for the bootstrap test [47].

### 4.3. Exon/Intron Organization and Motifs

To produce the diagram of the organization of exons/introns, we employed an online analysis on the Gene Structure Display Server (GSDS; http://gsds.cbi.pku.edu.cn) [48]. For this operation, we used the BLAST method of both CDS sequences and the genome sequences that we got from the PIECE 2 database http://www.bioinfogenome.net/piece/search.php for all 12 *DGK* genes to produce the diagram, before proceeding to the comparison of the intron/exon structures and gene evolution [49].

### 4.4. Proteins Domain Analysis and the Phylogenetic Relationship of GmDGK Proteins

To establish the multiple sequence alignments of all plant *DGK* genes, which were downloaded from the Phytozome protein database (https://phytozome.jgi.doe.gov/pz/portal.html), we used the DNAMAN program, and then we used the DOG 2.0 software [50] to draw the schematic diagram of the functional motifs in all soybean *DGK* genes. The MEGA6.0 software was used to align *DGK* gene sequences of different plant species [51]. By using the neighbor-joining (NJ) method of MEGA6.0, we created a phylogenetic tree, and we applied a bootstrap test with 1000 copies for each node [52].

### 4.5. Predictions of Subcellular Localization and Promoter Element Analysis for Soybean DGK Genes

We downloaded all soybean *DGK* gene sequences from Phytozome V11.0 and we BLASTed on PLACE 2 to find the database (https://sogo.dna.affrc.go.jp/cgi-bin/sogo.cgi) [53]. We also BLASTed the first 1500-bp upstream sequences of their transcription initiation sites in the PlantCare online tool (http://bioinformatics.psb.ugent.be/webtools/plantcare/html/) [54]. The presence of various cis-elements in the promoter regions of *GmDGK*s were predicted. Then, for the prediction of DGK protein subcellular locations in soybean, the DeepLoc-1.0 online tool (set for eukaryote cells) (http://www.cbs.dtu.dk/services/DeepLoc/) was used. We BLASTed a FASTA field containing all the protein sequences of both *Arabidopsis* and soybean DGKs and we ranged the results by cluster (Appendix A).

### 4.6. Plant Materials and Stress Treatments

*Glycine max* (Williams 82) was chosen as a model plant for soybean, and good seeds were selected and grown in hydroponic pots to ensure better growth. Hoagland’s solution (235 ppm K, 200 ppm Ca, 0.5 ppm B 31 ppm P, 64 ppm S, 48 ppm Mg, 0.5 ppm Mn, 0.05 ppm Zn, 0.02 ppm Cu, 210 ppm N, 0.01 ppm Mo, and 1 to 5 ppm Fe) was made to serve as a nutrient source. The plants were constantly kept under a 12-h light/12-h dark cycle, at 25 °C with 70% humidity, and the nutrient solution was replaced every 48 h. After 14 days, when the plants reached the four-leaf stage, we divided them into five different groups, then we transferred four of them into the various stress solutions, PEG (8% PEG8000), alkali (100 mM NaHCO_3_), salt (110 mM NaCl), and saline + alkali (70 mM NaCl + 50 mM NaHCO_3_) for 0, 1, 3, 6, 9, 12, and 24 h. The fifth group was kept growing in Hoagland’s solution without any stress and was used as the plant control [54]. Finally, the collection of the samples (roots and leaves) was done at around 11:00 a.m. and they were preserved at −80 °C for further usage.

To compare the tissue-specific expression of the *DGK* gene family in soybean, we used the untreated plant tissues as the samples (at 0 h).

### 4.7. RNA Extraction, Complementary DNA Synthesis, and Expression Analysis of Selected Soybean DGK Genes

After the collection of different samples (roots and leaves), total RNA from all collected tissues was separately extracted. We used the Trizol reagent (Invitrogen, Carlsbad, CA, USA) following the manufacturer’s protocol; then, after each RNA extraction, we checked the concentration and quality of the complementary DNA (cDNA) using a NanoDrop 2000 (ThermoFisher Scientific, Beijing, China). Afterward, each cDNA was integrated from 2 μg of the comparing RNA utilizing the PrimeScript RT reagent unit (Takara, Dalian, China). In total, more than 320 samples of soybean RNA were extracted and carefully preserved at −80 °C.

### 4.8. Quantitative Real-Time PCR Analysis

This study started with the selection of the reference genes for the qRT-PCR; *Actin11* (leaves) and *EF1A* (roots) were used as the reference genes, and the primers were designed for all *GmDGK*s. We used the Primer Premier 5 software to design all 24 primers as shown in Appendix A [55]. For expression analysis of the *GmDGK* gene under abiotic stresses, we performed RT-PCR on a Stratagene Mx3000P thermocycler (Agilent) using SYBR Green qPCR kits (Takara) with the following settings: 95 °C for 15 s; 40 cycles of 95 °C for 15 s; and annealing at 58 °C for 30 s. Each reaction was performed in triplicate, and three biological replicates were done in the growth chamber. The expression levels of each gene at different time stress points was calculated using the 2^−ΔΔCt^ method for abiotic treatments and the 2^ΔCt^ method for different tissues [56].

## 5. Conclusions

In conclusion, this study focused on the comprehensive genomic analysis and expression profiling of the diacylglycerol kinase (*DGK*) gene family in soybean (*Glycine max*) under abiotic stress. We identified 12 *DGK* genes in the whole genome of the soybean, which were distributed on six chromosomes. The subcellular location of their proteins was mostly predicted in the plasma membrane, the nucleus, and the cytoplasm, which are the key organelles of eukaryote cells. The conserved domains and phylogenetic analysis confirmed similarities and evolutionary relationships among soybean and other plant *DGK* genes, and the qRT-PCR results revealed their expression levels under various stress treatments in both leaf and root tissues. The results of the present work indicated that *GmDGK*s might be involved in soybean metabolism in response to abiotic stress. This study can enhance our knowledge of the roles of the *DGK* gene family in the soybean abiotic stress response.

## Figures and Tables

**Figure 1 ijms-20-01361-f001:**
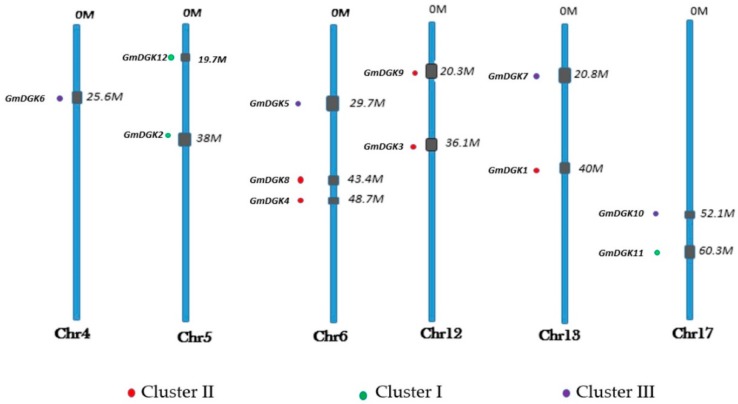
The stance of diacylglycerol kinase (*DGK*) genes family members on soybean chromosomes. Clusters are designated using separately colored spots.

**Figure 2 ijms-20-01361-f002:**
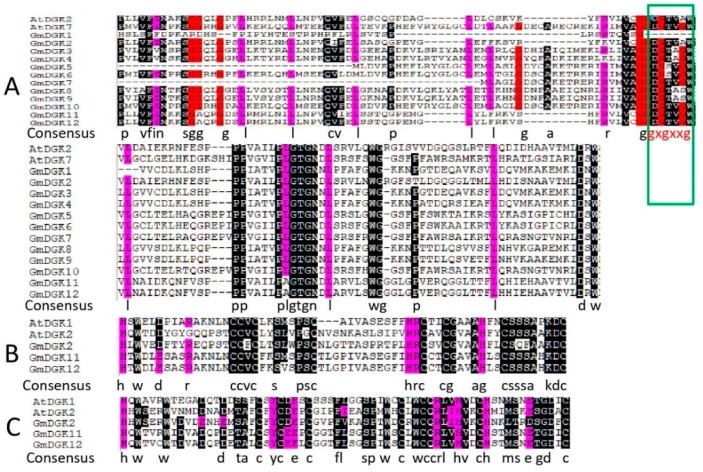
Multiple alignments of DGK catalytic (DGKc) domain (**A**), diacylglycerol/phorbol ester (DAG/PE)-binding domain C1 (**B**), and DAG/PE-binding domain 2 (**C**) in *Glycine max* and *Arabidopsis thaliana*.

**Figure 3 ijms-20-01361-f003:**
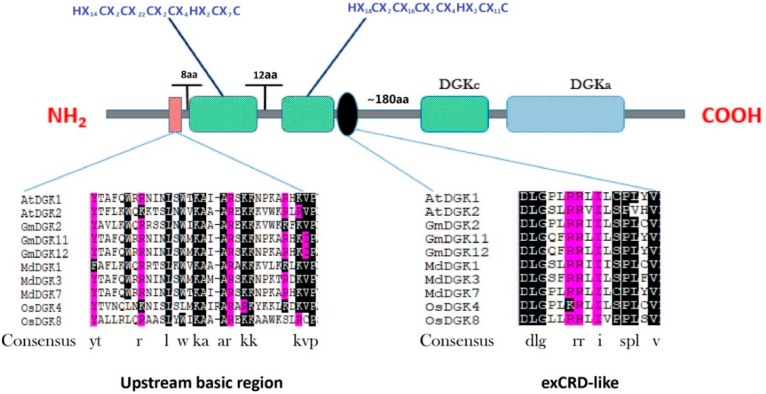
A detailed view of domains of *DGK* genes in cluster I with the predicted location and the sequences of conserved C_6_/H_2_ cores; the extended cysteine-rich domain (extCRD)-like domain and the upstream basic regions are also shown.

**Figure 4 ijms-20-01361-f004:**
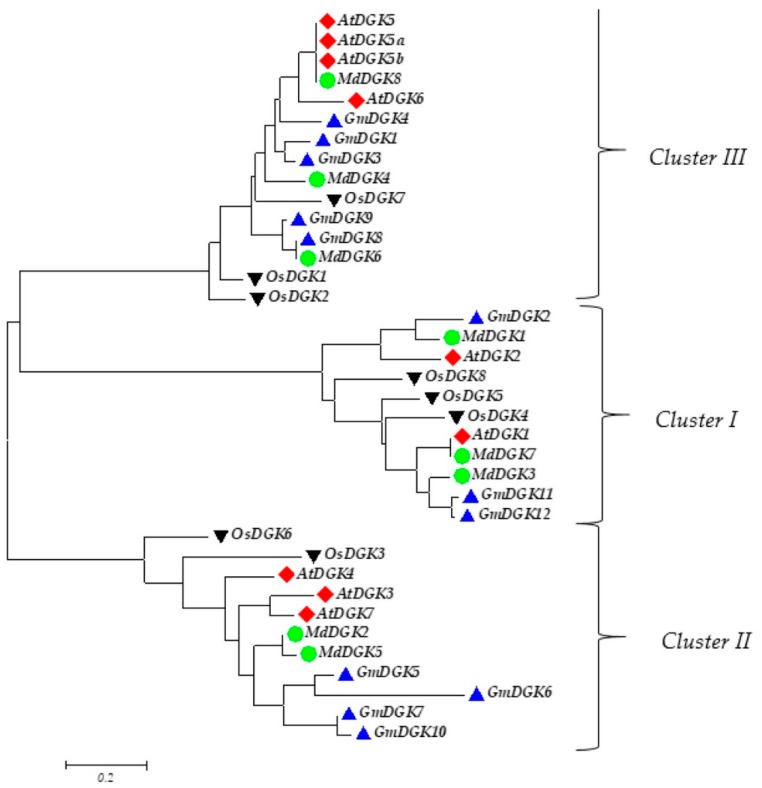
Phylogenetic analyses of *DGK* genes in soybean (Gm), *Arabidopsis* (At), rice (Os), and apple.

**Figure 5 ijms-20-01361-f005:**
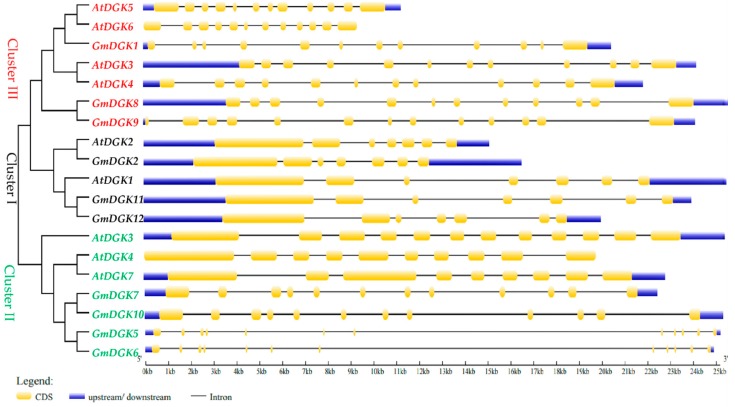
Phylogenetic intron/extron analysis, of the *Arabidopsis* and soybean *DGK* genes. Blue boxes represent upstream/downstream regions, gray lines represent introns, and yellow boxes represent exons.

**Figure 6 ijms-20-01361-f006:**
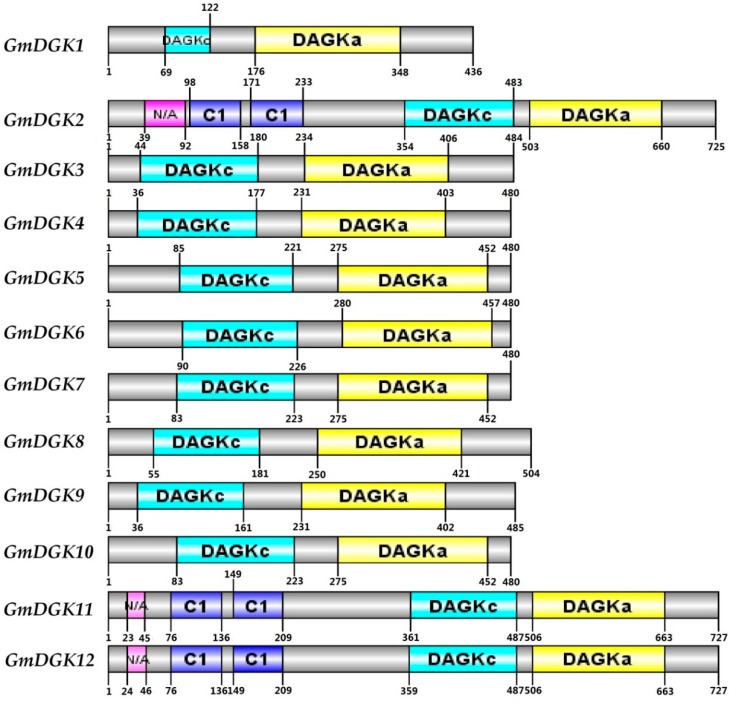
Functional domain analysis of soybean DGK proteins. The numbers show the positions of each amino acid in the protein.

**Figure 7 ijms-20-01361-f007:**
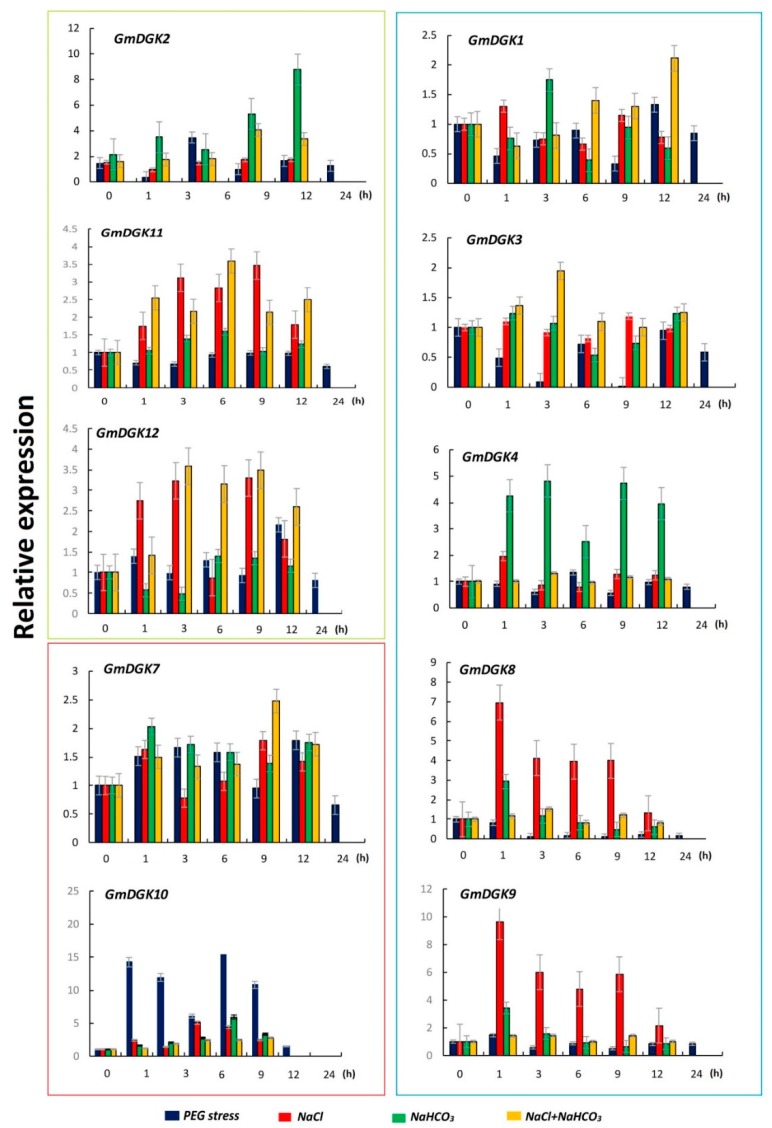
Soybean expression levels of *DGK* genes in leaf tissues under abiotic stresses with *Actin11* as a reference gene; the detection of their expression levels under different stress conditions at different time points is shown.

**Figure 8 ijms-20-01361-f008:**
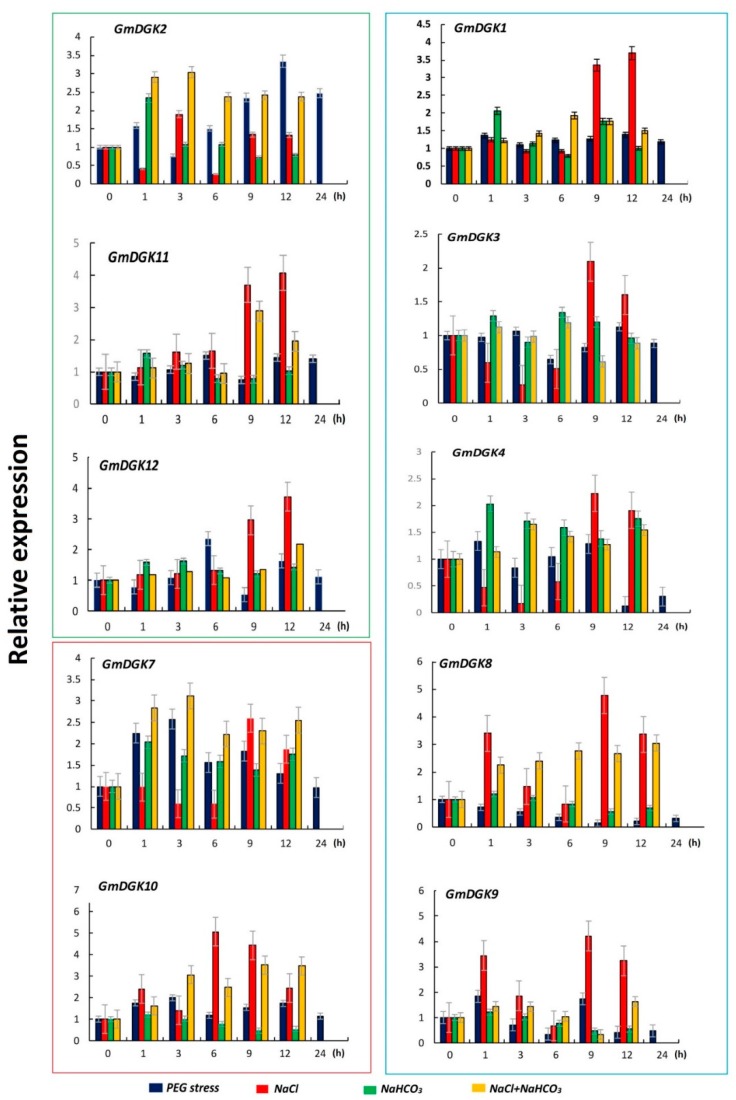
Soybean expression levels of *DGK* genes in root tissues under abiotic stresses with *EF1A* as a reference gene; the detection of their expression levels under various stress applications at different time points is shown.

**Figure 9 ijms-20-01361-f009:**
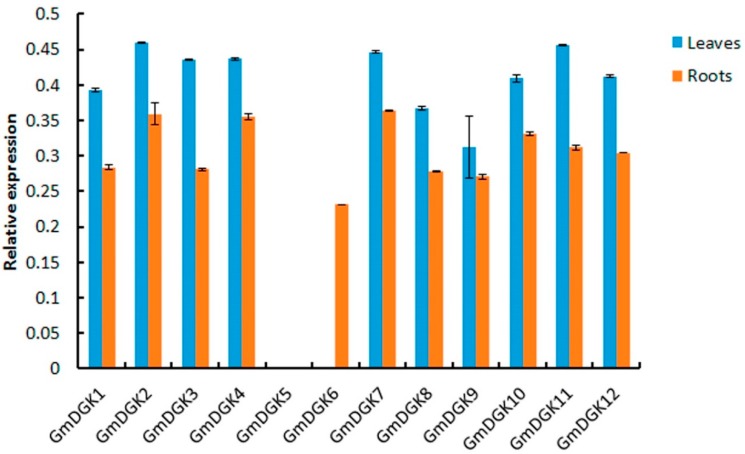
Tissue-specific expression values of *DGK* genes upon comparing qRT-PCR results in leaf (blue bands) and root (red bands) tissues of soybean, with *Actin11* as a reference gene for leaves and *EF1A* as a reference gene for roots.

**Figure 10 ijms-20-01361-f010:**
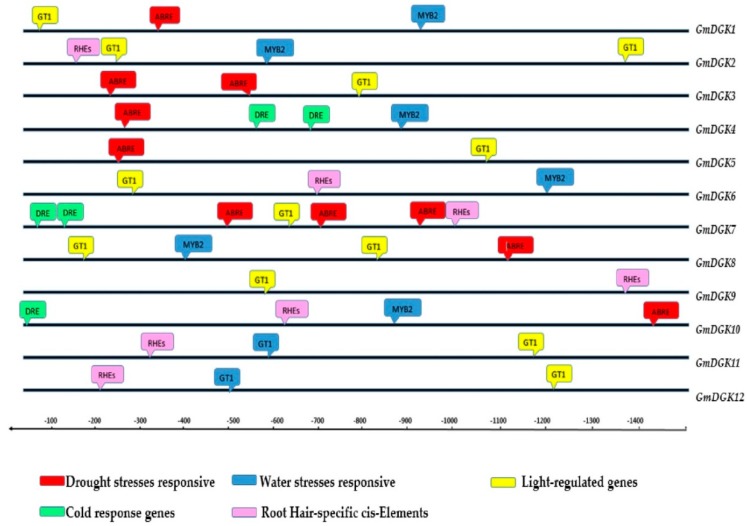
Illustration of five selected cis-regulatory units in all promoters of the *DGK* gene in soybean. From the translation start site, the upstream region from −1500 to −100 shows the upstream region of each *DGK* promoter. Several units including *ABRE*, *GT1*, *RHe*, *MYB2*, and *DRE* all exist in the promoter.

**Table 1 ijms-20-01361-t001:** Characterization of the soybean (*Glycine Max*) genome data transcript identifiers (IDs) with gene names, gene loci, gene identifiers, chromosomes (Chr), localization coordinates, ORF length (bp), isoelectric point (pI), protein identifiers, protein lengths (amino acids; aa), and protein weights (kDa).

Name	Gene Locus	Gene Identifier	Chr	Locations	Length (bp)	Proteins			
						Protein ID	Length (aa)	pI	(kDa)
*GmDGK1*	LOC100813110	Glyma.13G302200	13	39928623-39934694	1185	XP 003541921.1	436	6.77	48.62
*GmDGK2*	LOC100797649	Glyma.05G196100	5	38025457-38031403	2178	XP_006580353.1	725	7.05	78.6
*GmDGK3*	LOC100813809	Glyma.12G200100	12	36142043-36148329	1467	XP_014632628	480	5.59	53.7
*GmDGK4*	LOC100817198	Glyma.06G299200	6	48769786-48776248	1456	XP_003526295	484	5.29	54.29
*GmDGK5*	LOC100811505	Glyma.06G223900	6	29792271-29817479	1443	XP_014632628	480	6.36	53.76
*GmDGK6*	LOC100794923	Glyma.04G143000	4	25618714-25643610	1458	XP_003552048	485	6.36	54.2
*GmDGK7*	LOC100811680	Glyma.13G093100	13	20812774-20820984	1443	XP_003542286	480	7.61	53.54
*GmDGK8*	LOC100816144	Glyma.06G254900	6	43416272-43423748	1452	XP_014632201	504	7.29	56.32
*GmDGK9*	LOC100801004	Glyma.12G146700	12	20308574-20316372	2044	XP_006592585	485	6.58	54.35
*GmDGK10*	LOC100808353	Glyma.17G067400	17	5211623-5219147	1874	XP_014625529.3	480	7.46	53.56
*GmDGK11*	LOC100794891	Glyma.17G077100	17	6032535-6039406	2184	XP_014625346.1	727	6.04	80.97
*GmDGK12*	LOC100802426	Glyma.05G022500	5	1971267-1978461	2184	XP_006579516	727	6	80.93

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
