# Peer review of "Comprehensive Genomic Analysis and Expression Profiling of Diacylglycerol Kinase (DGK) Gene Family in Soybean (Glycine max) under Abiotic Stresses"

_ijms, 2019, doi:10.3390/ijms20061361_

Reviewer 1 Report

The manuscript, Comprehensive genomic analysis and expression profiling of DGK gene family in soybean (Glycine max) under abiotic stresses by Carther et al., is well presented. 

However, I have a few major concerns:

The manuscript deal with the research question; in silico identification, and elucidation of the role of the Diacylglycerol kinase genes in G. max. Authors should give an introductory figure depicting the function of DGKs in general. Write the full form of DGK in the manuscript title.     

Section 4.5. Predictions of subcellular localization, the authors did not mention how did they predict the subcellular localization of DGK genes in G.max. The authors did not discuss these results at all. The predictions show most of the GmDGK genes are nuclear located (Tabel S3), how authors summarize these results with the functions of these genes, where are corresponding Arabidopsis DGK gene orthologues (shown in Figure 5) are localized in the cell?

The discussion is very subjective. The authors did not justify their results in context with the hypothesis. Particularly about the gene expression patterns of the twelve DGKs. Line 301-303 where the authors are claiming that all soybean DGKs could contain ABREs in their promoters, however, in their data analysis they have shown the predicted ABREs in only 1,3,4,5,8, and 10 DGKs (Fig. 10). Moreover, they have given the reference of  Cacas et al., 2017 Reference No-33 which I think irrelevant. To get more out of the gene expression data to understand the role of GmDGKs in abiotic stress, I suggest the authors present the qRT-PCR data for each treatment separately for the genes in each cluster. Also, I wondered why the authors did not use ABA treatment to examine the stress responsiveness of these genes. I strongly encourage to perform the ABA treatment which should answer this question.     

Line 2019-220, the authors said that- This suggests that the DGK enzymes could play an interesting role in the PA pathway during roots and leaves development (20). What does an INTERESTING mean? What does the Reference 20 signify? I found that this reference Han et al., 2011 is irrelevant. 

Conclusion section is poorly written. In Line 385-386, did you mean DGK genes in the whole genome of the soybean were distributed on 6 of the 20 soybean chromosomes? Repetitive sentences. 

Which data they used for Figure 9. 

Supplementary Table S3, when clicked on the details, the URL did not work for more information. The values presented in red color, what does this mean? 

Line 74, Glycin max should be italic. 

Line 217 and 232- should be Figure 9 

Section 4.8, Line 374- the sentence needs to be corrected

Reference section: inconsistency in reference format  

Overall, the authors have invested in answering a good research question; however, the manuscript lacks scientific soundness, proper referencing and comprehensive introduction, discussion and conclusion. I encourage authors to re-write the manuscript.

Author Response

Thank you for your comments. We have improved the overall English language of the manuscript. Our answers to your points are as follows.

Point 1: The manuscript deal with the research question; in silico identification, and elucidation of the role of the Diacylglycerol kinase genes in G. max. Authors should give an introductory figure depicting the function of DGKs in general. Write the full form of DGK in the manuscript title.

Response 1: We agree with the reviewer and have added an introductory figure (Supplement Fig. 1) depicting the function of DGKs in general, and a brief definition is now added at the introduction section (line 368-374) in the manuscript. Thanks

Point 2: Section 4.5. Predictions of subcellular localization, the authors did not mention how did they predict the subcellular localization of DGK genes in G.max? The authors did not discuss these results at all. The predictions show most of the GmDGK genes are nuclear located (Table S3), how authors summarize these results with the functions of these genes, where are corresponding Arabidopsis DGK gene orthologues (shown in Figure 5) are localized in the cell?

 Response 2: We agree with the reviewer and thank you for your suggestions.

We have analysed and provided new data on the prediction of the DGKs subcellular localization for both soybean and Arabidopsis, using the blasting method in the DeepLoc1.0 online tool (http://www.cbs.dtu.dk/services/DeepLoc/), and added the sentences emphasizing this point in the results section (Line 335 – 341), discussion (Line 434 – 449) and material and methods section (Line 547 –551). Then we ranged data into different groups relating to their clusters (supplement Table. S3)

 Point 3: The discussion is very subjective. The authors did not justify their results in context with the hypothesis. Particularly about the gene expression patterns of the twelve DGKs. Line 301-303 where the authors are claiming that all soybean DGKs could contain ABREs in their promoters, however, in their data analysis they have shown the predicted ABREs in only 1, 3,4,5,8 and 10 DGKs (Fig. 10). Moreover, they have given the reference of Cacas et al., 2017 Reference No-33 which I think irrelevant. To get more out of the gene expression data to understand the role of GmDGKs in abiotic stress, I suggest the authors present the qRT-PCR data for each treatment separately for the genes in each cluster. Also, I wondered why the authors did not use ABA treatment to examine the stress responsiveness of these genes. I strongly encourage to perform the ABA treatment which should answer this question.      

 Response 3: Thank you. As suggested, we have reorganized qRT-PCR data for each treatment separately for the genes in each cluster. (Fig. 8 and 9), and sentences emphasizing the new data arrangement are added in the results section (Line 265 - 294.) and discussion (Line 422 - 453).

 We have rectified the mistake about the ABREs contain of soybean DGK genes in the line (448 - 452)

 Point 4: Line 2019-220, the authors said that- This suggests that the DGK enzymes could play an interesting role in the PA pathway during roots and leaves development (20). What does an INTERESTING mean? What does the Reference 20 signify? I found that this reference Han et al., 2011 is irrelevant.

 Response 4: We agree with the reviewer and we have provided precisions about the purpose used method, and the expression of DGK genes in roots and leaves tissues in the section results. Line (297 – 299) and (306 - 309), Thank you for your suggestions.

 Point 5: Conclusion section is poorly written. In Line 385-386, did you mean DGK genes in the whole genome of the soybean were distributed on 6 of the 20 soybean chromosomes? Repetitive sentences.

 Response 5: We agree with the reviewer and have revised the conclusion. Thank you.

Line. 598 - 608

“In conclusion, this study identified 12 DGK genes in the whole genome of the soybean, widely distributed on the soybean chromosomes, and which their proteins were located in the key organelles of the eukaryote cell. We also pointed the potential roles as the transmembrane proteins for the cluster I DGKs in soybean. While the conserved domain and phylogenetic analysis confirmed similarities and evolutionary relationships among soybean and other plant DGK genes. The qRT-PCR results bring out curious candidates for further exploration of their functional activities such as GmDGK2, GmDGK4, GmDGK7, and GmDGK10. The results of the present work indicated that the GmDGKs could play a crucial role in soybean stress responses, and will enhance our knowledge of the roles of the DGK genes family in the soybean abiotic stress response.”

 Point 6: Which data they used for Figure 9.

 Response 6: The data used for the figure 9 (Figure 7) were collected in none stressed leaf/root and calculated according to the 2ΔCt method.

Line 584 - 587

 Point 7: Supplementary Table S3, when clicked on the details, the URL did not work for more information. The values presented in red color, what does this mean?    

 Response 2: We agree with the reviewer and thank you for your suggestions.

We have analysed and provided new data on the prediction of the DGKs subcellular localization for both soybean and Arabidopsis, using the blasting method in the DeepLoc1.0 online tool, (http://www.cbs.dtu.dk/services/DeepLoc/) and added sentences emphasizing this point in the material and methods section (Line 547 –551), results section (Line 334 – 340) and discussion (Line 441 – 447). Then we ranged data into different groups relating to their clusters (supplement Table. S3)

 Point 8: Line 74, Glycine max should be italic.

 Response 8: done

Point 9: Line 217 and 232- should be Figure 9

 Response 9: done

Point 10: Section 4.8, Line 374- the sentence needs to be corrected

 Response 10: done

Point 11: Reference section: inconsistency in reference format 

 Response 11:

Overall, the authors have invested in answering a good research question; however, the manuscript lacks scientific soundness, proper referencing and comprehensive introduction, discussion and conclusion. I encourage authors to re-write the manuscript.

 We agree with the reviewer and we have rewritten a new introduction (38 - 91), a new discussion (line 361 – 453), and a new conclusion (line 598 - 606). This, according to the new data arrangement.

 Thanks for your great contribution.

Reviewer 2 Report

In the manuscript entitled ‘Comprehensive genomic analysis and expression 
profiling of DGK gene family in soybean (Glycine max) under abiotic stresses’’ by Kue Foka Idrice Carther 
and colleagues, the authors performed genome-wide analysis and expression profiling of DGK gene family in the soybean to characterize of DGK gene family and their importance in abiotic stress response. The work will surely contribute to the molecular understanding of overall plant responses to abiotic stresses as well. I suggest the authors to discuss the results obtained in gene expression analysis of the distinct DGKs showing remarkable changes under each stress condition in reference to the current literature. The references are also not arranged uniformly in the manuscript.

Author Response

In the manuscript entitled ‘Comprehensive genomic analysis and expression profiling of DGK gene family in soybean (Glycine max) under abiotic stresses’’ by Kue Foka Idrice Carther and colleagues, the authors performed genome-wide analysis and expression profiling of DGK gene family in the soybean to characterize of DGK gene family and their importance in abiotic stress response. The work will surely contribute to the molecular understanding of overall plant responses to abiotic stresses as well.

 Thank you for your comments. We have improved the overall English language of the manuscript. Our answers to your points are as follows.

 Point 1: I suggest the authors to discuss the results obtained in gene expression analysis of the distinct DGKs showing remarkable changes under each stress condition in reference to the current literature.

 Response 1: We agree with the reviewer and we have provided precisions into the gene expression analysis of the distinct DGKs, showing remarkable changes under each stress condition in reference to the current literature. As suggested, we have reorganized the qRT-PCR data for each treatment separately for the genes in each clusters. (Fig. 8 and 9), and sentences emphasizing these new data arrangements are added in the results section (Line 269 - 294.) and discussion (Line 422 - 453). Thank you.

 Point 2: The references are also not arranged uniformly in the manuscript.

 Response 2: thanks for your observations the references are now arranged uniformly in the manuscript.

 Thanks for your great contribution.

Reviewer 3 Report

Manuscript „comprehensive genomic analysis and expression profiling of DGK gene family in soybean (Glycine  max) under abiotic stresses” contains detailed bioinformatics analysis of DGK gene family supplemented with expression experiment based on qPCR. Significance of the research is well justified. Results are presented clearly in logical order. Some fragments need language correction. Description of Fig.9 “gotten by all … RNA-Seq)” needs to be corrected or some explanation provided in methods section. Selection of Wiliams 82 as model plant would be better justified if genotype was sequenced of is recognized as stress tolerant -  these data should be added if available.

Author Response

Manuscript “comprehensive genomic analysis and expression profiling of DGK gene family in soybean (Glycine max) under abiotic stresses” contains detailed bioinformatics analysis of DGK gene family supplemented with expression experiment based on qPCR. Significance of the research is well justified. Results are presented clearly in logical order. Some fragments need language correction.

 Thank you for your comments. We have improved the overall English language of the manuscript. Our answers to your points are as follows.

  Point 1: Description of Fig.9 “gotten by all … RNA-Seq)” needs to be corrected or some explanation provided in methods section. Selection of Williams 82 as model plant would be better justified if genotype was sequenced of is recognized as stress tolerant - these data should be added if available.

 Response 1: We agree with the reviewer and we have provided precisions on the purpose for using the soybean William 82 as model plant, and the paragraph emphasizing that dada is added in the discussion section (line 385 – 388)

 Line 385 – 388 “This curiosity prompted our study concerning the roles of DGK genes family in soybean, and for this study, we chose the soybean William 82 as model, because of three factors: Its whole genome had been sequenced in previous study [15], it is identified as having an efficient drought-response tolerance [30] and its tolerance could be improved [31].“

 Thanks for your great contribution.

Round  2

Reviewer 1 Report

Major Comments:

The authors have taken a sincere note of all my comments. 

However, the manuscript still sounds like someone's thesis writeup, there is still great scope to improve the introduction, and mainly the conclusion. 

Rewrite the conclusion. 

Extensive English editing is required. The authors must avoid words such as crowned, tiny, surgery, curious candidates etc., must be avoided. For example, Lines 83- 84:  Then, we crowned the 84 bioinformatics analysis by monitoring the real-time PCR....   

Minor suggestions:

Line 38: Drought stresses, did you mean- Drought stress? 

Lines-87-88:  Thus, we have realized a genome-wide identification and analysis of DGK gene family members in Glycine max...what does it mean?

Line 301- Page 13- tiny expression of the GmDGK6 in the root??? 

Lines 27, page 568 & 578- jealously preserved at −80°C for further usage???

Line  580: This surgery started with the selection of the reference

Lines 265-266: A qRT-PCR was monitoring to recognize all the GmDGKs transcript levels

Did you mean the qRT-PCR analysis was performed to monitor GmDGKs transcript levels?

Line 149 page-5- did you mean DESIGNATED?   

Lines 274- 276- In general most of the DGK 275 genes expressions were up and down regulated in both roots and leave tissues throughout the PEG 276 stress application 

This is a vague sentence, do not make any conclusion.   

Lines 290-292- rephrase the sentence 

Supplementary Figure 1: phosphate acid (PA) must be Phosphatidic acid

Lines 541-542: The alignment of DGK 542 gene sequences of different plants species was also realized by MEGA6.0 software???

Author Response

Point 1. Extensive English editing is required. The authors must avoid words such as crowned, tiny, surgery, curious candidates etc., must be avoided. For example, Lines 83- 84:  Then, we crowned the 84 bioinformatics analysis by monitoring the real-time PCR....   Response 1. Thank you for your comments. We have improved the overall English language of the manuscript. Our answers to your points are as follows. Line 38: Drought stress represents the most important limiting factor that affect Lines-87-88:  Thus, we have realized a genome-wide identification and analysis of DGK gene family members in Glycine max...what does it mean? Line 93: We replaced “realized” with an appropriate word “done” Line 301- Page 13- tiny expression of the GmDGK6 in the root??? Line 336: (With a very low expression of the GmDGK6 in the root) Lines 27, page 568 & 578- jealously preserved at −80°C for further usage??? Line 451: In total more than 320 samples of soybean RNA were extracted and carefully preserved at -80 C. Line 580: This surgery started with the selection of the reference Line 454: This study started with the selection of the reference genes for the qRT-PCR Lines 265-266: A qRT-PCR was monitoring to recognize all the GmDGKs transcript levels Did you mean the qRT-PCR analysis was performed to monitor GmDGKs transcript levels? Lines 199: The qRT-PCR analysis was performed to monitor GmDGKs transcripts level levels in leaves and roots tissues under PEG, saline, alkaline and saline/alkaline treatments. Line 149 page-5- did you mean DESIGNATED?   Line 102: We identified a total of twelve DGK genes that we designated as GmDGK (1-12) Lines 274- 276- In general most of the DGK 275 genes expressions were up and down regulated in both roots and leave tissues throughout the PEG 276 stress application This is a vague sentence, do not make any conclusion.   Deleted Lines 290-292- rephrase the sentence Lines 225-227: Similar study was conducted on maize (Zea mays) and reported that the relative expression levels of ZmDGK7 were prominent in many tissues under salt stress treatments [16]. Supplementary Figure 1: phosphate acid (PA) must be Phosphatidic acid Done Lines 541-542: The alignment of DGK 542 gene sequences of different plants species was also realized by MEGA6.0 software??? Lines 416-417: The MEGA6.0 software was used to align DGK gene sequences of different plants species [51]. Point 2. Rewrite the conclusion. Response 2. Line 466-476: In conclusion, this study was focused on the Comprehensive genomic analysis and expression profiling of Diacylglycerol kinase (DGK) gene family in soybean (Glycine max) under abiotic stresses. We have identified 12 DGK genes in the whole genome of the soybean which were distributed on six chromosomes in the soybean genome. The subcellular location of their proteins were mostly predicted in the plasma membrane, the nucleus and the cytoplasm which are the key organelles of the eukaryote cells. The conserved domain and phylogenetic analysis confirmed similarities and evolutionary relationships among soybean and other plants DGK genes, and the qRT-PCR results revealed their expression pattern levels under various stress treatments in both leaves and roots tissues. The results of the present work indicated that the GmDGKs might be involved in soybean metabolisms in response to abiotic stress. This study will enhance our knowledge of the roles of the DGK genes family in the soybean abiotic stress response. Thanks for your great contribution.
